# Evolutionary plasticity of SH3 domain binding by Nef proteins of the HIV-1/SIVcpz lentiviral lineage

Zhe Zhao[1], Riku Fagerlund[1], Helena Tossavainen[2], Kristina Hopfensperger[3], Rishikesh Lotke[3], Smitha Srinivasachar Badarinarayan[3], Frank Kirchhoff[4], Perttu Permi[2,5], Kei Sato[6], Daniel Sauter[3,4], Kalle Saksela[1] *

1 Department of Virology, University of Helsinki and Helsinki University Hospital, Helsinki, Finland,
2 Department of Biological and Environmental Science, University of Jyväskylä, Jyväskylä, Finland,
3 Institute for Medical Virology and Epidemiology of Viral Diseases, University Hospital Tübingen, Tübingen, Germany, 4 Institute of Molecular Virology, Ulm University Medical Center, Ulm, Germany, 5 Department of Chemistry, Nanoscience Center, University of Jyväskylä, Jyväskylä, Finland, 6 Division of Systems Virology, Department of Infectious Disease Control, International Research Center for Infectious Diseases, Institute of Medical Science, The University of Tokyo, Tokyo, Japan

* kalle.saksela@helsinki.fi

**Data Availability Statement:** All relevant data are within the manuscript and its Supporting Information files.

## Abstract

The accessory protein Nef of human and simian immunodeficiency viruses (HIV and SIV) is an important pathogenicity factor known to interact with cellular protein kinases and other signaling proteins. A canonical SH3 domain binding motif in Nef is required for most of these interactions. For example, HIV-1 Nef activates the tyrosine kinase Hck by tightly binding to its SH3 domain. An archetypal contact between a negatively charged SH3 residue and a highly conserved arginine in Nef (Arg77) plays a key role here. Combining structural analyses with functional assays, we here show that Nef proteins have also developed a distinct structural strategy—termed the "R-clamp"—that favors the formation of this salt bridge via buttressing Arg77. Comparison of evolutionarily diverse Nef proteins revealed that several distinct R-clamps have evolved that are functionally equivalent but differ in the side chain compositions of Nef residues 83 and 120. Whereas a similar R-clamp design is shared by Nef proteins of HIV-1 groups M, O, and P, as well as SIVgor, the Nef proteins of SIV from the Eastern chimpanzee subspecies (SIVcpz[P.t.s.]) exclusively utilize another type of R-clamp. By contrast, SIV of Central chimpanzees (SIVcpz[P.t.t.]) and HIV-1 group N strains show more heterogenous R-clamp design principles, including a non-functional evolutionary intermediate of the aforementioned two classes. These data add to our understanding of the structural basis of SH3 binding and kinase deregulation by Nef, and provide an interesting example of primate lentiviral protein evolution.

## Author summary

Viral replication depends on interactions with a plethora of host cell proteins. Cellular protein interactions are typically mediated by specialized binding modules, such as the

**Funding:** KS was funded by the Helsinki University Central Hospital Research Council grant TYH2017248 and by the Jane and Aatos Erkko Foundation (JAES2016). PP was supported by grants from the Academy of Finland (323435) and Jane and Aatos Erkko Foundation (JAES2019). DS was supported by the Heisenberg Program (SA 2676/3-1) and the Priority Program SPP1923 (SA 2676/1-2) of the German Research Foundation (DFG). The funders had no role in study design, data collection and analysis, decision to publish, or preparation of the manuscript.

**Competing interests:** The authors have declared that no competing interests exist.

SH3 domain. To gain access to host cell regulation viruses have evolved to contain SH3 domain binding sites in their proteins, a notable example of which is the HIV-1 Nef protein. Here we show that during the primate lentivirus evolution the structural strategy that underlies the avid binding of Nef to cellular SH3 domains, which we have dubbed the R-clamp, has been generated via alternative but functionally interchangeable molecular designs. These patterns of SH3 recognition depend on the amino acid combinations at the positions corresponding to residues 83 and 120 in the consensus HIV-1 Nef sequence, and are distinctly different in Nef proteins from SIVs of Eastern and Central chimpanzees, gorillas, and the four groups of HIV-1 that have independently originated from the latter two. These results highlight the evolutionary plasticity of viral proteins, and have implications on therapeutic development aiming to interfere with SH3 binding of Nef.

## Introduction

Primate lentiviruses comprise human immunodeficiency viruses (HIV-1 and HIV-2), as well as more than 40 simian immunodeficiency viruses (SIVs) infecting non-human primate species from sub-Saharan Africa. HIV-1 groups M, N, O, and P originate from four independent transmissions of SIV from Central chimpanzees (SIVcpz$^{\text{P.t.t.}}$) and gorillas (SIVgor) to humans, whereas HIV-2 groups A to I are derived from nine zoonotic transmissions of SIV from sooty mangabeys (SIVsmm) (reviewed in [1]). SIVsmm was also transmitted to macaques, giving rise to SIVmac. The global AIDS pandemic is largely caused by HIV-1 group M strains, which are further divided into 10 subtypes (A, B, C, D, F, G, H, J K, and L), as well as several sub-subtypes and circulating recombinant forms [2–4].

The viral protein Nef is a multifunctional accessory factor encoded by all primate lentiviruses. While *nef*-defective HIV and SIV are replication-competent, infections with such viruses are associated with low viral loads and no or delayed pathogenesis in humans or experimentally infected macaques [5]. Nef itself exhibits no enzymatic activity but instead modulates host cell function by interacting with a plethora of other proteins to hijack cellular signaling and membrane trafficking pathways [6–8].

A function of HIV-1 Nef that is well understood at the molecular and mechanistic level is its interaction with Hck, a member of the Src-family tyrosine kinases. Nef tightly binds to the SH3 domain of Hck, thereby shifting it from an intramolecular autoinhibitory state into a catalytically active conformation [9,10]. Hck plays an important role in regulating activation and effector functions of macrophages, the second main target cell population of HIV-1 besides CD4$^+$ T cells [11,12]. Many Nef functions in other cell types including T lymphocytes, (e.g., interfering with Lck localization and immunological synapse formation [13], and reorganization of the actin cytoskeleton [14,15]) are also strictly dependent on its SH3 binding capacity. However, the relevant SH3 proteins in these cells do not bind to Nef as tightly as Hck, and have therefore remained less well characterized [7].

The discovery of the Nef-Hck interaction provided the first example of pathogen takeover of host cell signaling via SH3 domain ligand mimicry [16]. Src Homology 3 (SH3) domains are short (~60 aa) modular protein units specialized for mediating protein interactions via proline-rich core binding sites in the target proteins, and are ubiquitous (~300 in human) in eukaryotic proteins involved in regulation of cell behavior [17,18].

Studies on the Nef-SH3 complex also revealed a new paradigm for the structural basis of SH3 binding: It could be shown that docking of the proline-rich (PxxP) peptide of Nef by the SH3 core binding interface is assisted by further molecular contacts of the so-called SH3 RT-

loop region to provide additional specificity and affinity to this interaction [9]. The amino acid residues that form a binding pocket for the RT-loop of the Hck-SH3 domain are highly conserved in HIV-1 and HIV-1-like Nef proteins, and provide them with a capacity for strong binding to Hck [9,19]. Nef proteins from the HIV-2/SIVsmm groups lack these residues and hence strong Hck binding capacity, but as evidenced by the SIVmac293-Y113W/E117T/E118Q triple mutant [20], can acquire this function upon introduction of just a few key residues from HIV-1 Nef.

We have recently shown that Hck activation by Nef leads to Raf/MAPK pathway activation and triggers the secretion of proinflammatory cytokines [19,21]. In agreement with their enhanced affinity for Hck [9,20], we found that only HIV-1/SIVcpz type Nef proteins but not HIV-2/SIVsmm type Nef proteins show this function. The current study was incited by our observation that an HIV-1 group N Nef (clone YBF30) was unable to bind and activate Hck and the Raf/MAPK pathway for reasons that could not be readily explained by its amino acid sequence. Our investigation into this issue led to the discovery of a structural arrangement that we have termed the arginine (R)-clamp and characterized here. We show that different but functionally equivalent R-clamps have emerged in the HIV-1/SIVcpz lineage to coordinate Hck SH3 domain binding by Nef. This finding further highlights the importance of Nef-mediated SH3 binding and the enormous plasticity of primate lentiviral accessory proteins.

## Results

In our previous studies we examined the capacity of Nef to activate Hck by monitoring tyrosine phosphorylation of paxillin, a prominent substrate for Src family tyrosine kinases, and by measuring the induction of AP-1-regulated reporter gene expression following Hck-activated MAPK signaling [19,21]. When additional Nef proteins from HIV-1 groups N, O, and P were examined (Fig 1), we unexpectedly noticed that the Nef protein of HIV-1 N YBF30 (AJ006022) was unable to induce paxillin phosphorylation or AP-1 activity despite sharing the $^{72}$PxxPxR$^{77}$ motif (numbering based on the HIV-1 Nef consensus) known to be important for Hck binding by HIV-1/SIVcpz type Nefs (Fig 2). Importantly, YBF30 Nef was clearly expressed in these cells (Fig 1A), and as also shown previously [22] was functional when tested for its capacity to downregulate cell surface expression of CD4 (Figs 1C and S1).

To examine if this is a characteristic of several HIV-1 group N viruses we tested three additional HIV-1 N Nef proteins from the strains 02CM-DJO0131 (AY532635), YBF106 (AJ271370), and S4858 (KY498771). However, we found that all of them were fully competent for Hck activation, and induced paxillin phosphorylation and AP-1 activity as efficiently as Nef from the widely used HIV-1 group M strain SF2 (Fig 1).

Since the core SH3 docking motif ($^{72}$PxxPxR$^{77}$) as well as the key RT-loop accommodating residues, including F90, W113, T117, and Q118, were conserved in YBF30 Nef (see Fig 2) we had a more detailed look at the original X-ray structure of the HIV-1 Nef-SH3 complex (1EFN; [9]), which directed our attention to Nef residue 120. This position is occupied by an aromatic residue (tyrosine or phenylalanine) in virtually all HIV-1 M Nef proteins, while an isoleucine residue is found in this position in YBF30 Nef (Fig 2). In the 1EFN structure, Y120 can be seen to coordinate the Nef-SH3 interaction by buttressing the side-chain of the R77 in the Nef $^{72}$PxxPxR$^{77}$ motif together with the SH3 residue W37 (SH3 numbering according to [18]) in order to stabilize the close positioning of R77 with the acidic SH3 residue D17 to form a critical salt bridge with it (Fig 3A and 3B). We termed this steric guiding of the Nef residue R77 into close proximity of D17 in Hck-SH3 the "arginine (R)-clamp", and hypothesized that the failure of YBF30 Nef to activate Hck is due to the failure of its Ile120 residue to from a functional R-clamp (Fig 3C).

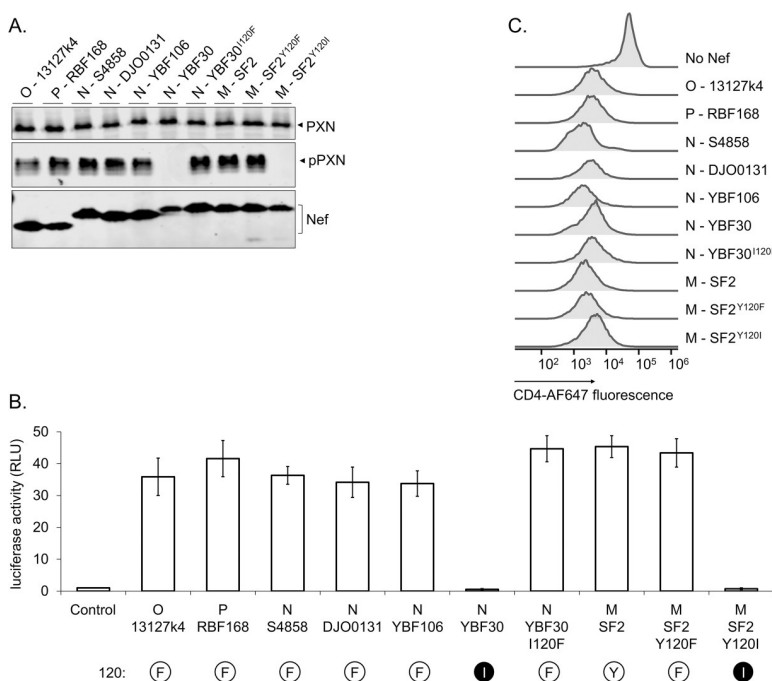

**Fig 1. Hck-activating potential of selected HIV-1 Nefs. (A)** Hck-expressing HZ-1 cells were co-transfected with vectors for paxillin and Nef from the indicated HIV-1 group M, N, O or P clones, or mutants thereof. Lysates of the transfected cells were analyzed by Western blotting using antibodies against paxillin (PXN), phosphorylated paxillin (pPXN) and Nef. **(B)** HZ-1 cells were transfected with an AP-1 transcription factor-driven luciferase reporter alone (Control) or together with the indicated Nefs variants. Luciferase activity was measured in cells harvested 24 h post-transfection, and normalized to the corresponding control sample that was set to 1. The data shown are derived from three independent experiments, with SE indicated by error bars. The amino acid at position 120 of each Nef protein is indicated, aromatic residues are shown as a black 1-letter symbol in a white sphere, and isoleucine as a white letter in a black sphere. **(C)** Jurkat T cells stably expressing CD4 were infected with lentiviral vectors expressing GFP alone (No Nef) or together with plasmids expressing the indicated Nefs. CD4 down-regulation by Nef was measured using flow cytometry 48 h after lentiviral transduction, and histograms illustrating cell surface levels of CD4 among the GFP positive Jurkat cells are shown. Original dot plots are shown in S1 Fig.

In support of this hypothesis, we found that YBF30 Nef became fully competent for Hck activation when its I120 residue was replaced by a phenylalanine (YBF30 I120F) (Fig 1). We then further investigated this idea by changing the Y120 residue in the Nef protein from the HIV-1 M laboratory strain SF2 into an YBF30-like isoleucine (Nef mutant SF2 Y120I) or into a phenylalanine, the other commonly found residue at this position (Nef mutant SF2 Y120F). While the conservative Y120F amino acid change did not affect Hck activation, the Y120I mutation recapitulated the failure of YBF30 Nef to stimulate paxillin phosphorylation and AP-1 activity, without substantially compromising protein expression levels or CD4 downregulation (Fig 1). Thus, we conclude that the lack of a functional R-clamp due to the I120 residue likely explains the inability of YBF30 Nef to activate Hck.

Analysis of additional HIV-1 group N Nefs revealed that in 5 out of the total of 12 sequences that are available in databases the position 120 was occupied by Ile (Table 1). This is in stark contrast to the Nef proteins from the three other HIV-1 groups M, O, and P, in which residue 120 is invariably Y or F. We had access to another I120-containing HIV-1 N Nef namely 2693BA (GQ925928), and went on to examine its ability to activate Hck. Having already established a strict correlation between paxillin phosphorylation and AP-1 activity as markers of Hck activation (Fig 1 and [19]), we chose to focus on using AP-1-driven reporter gene expression as the read-out in our further studies. Unexpectedly, we found that despite

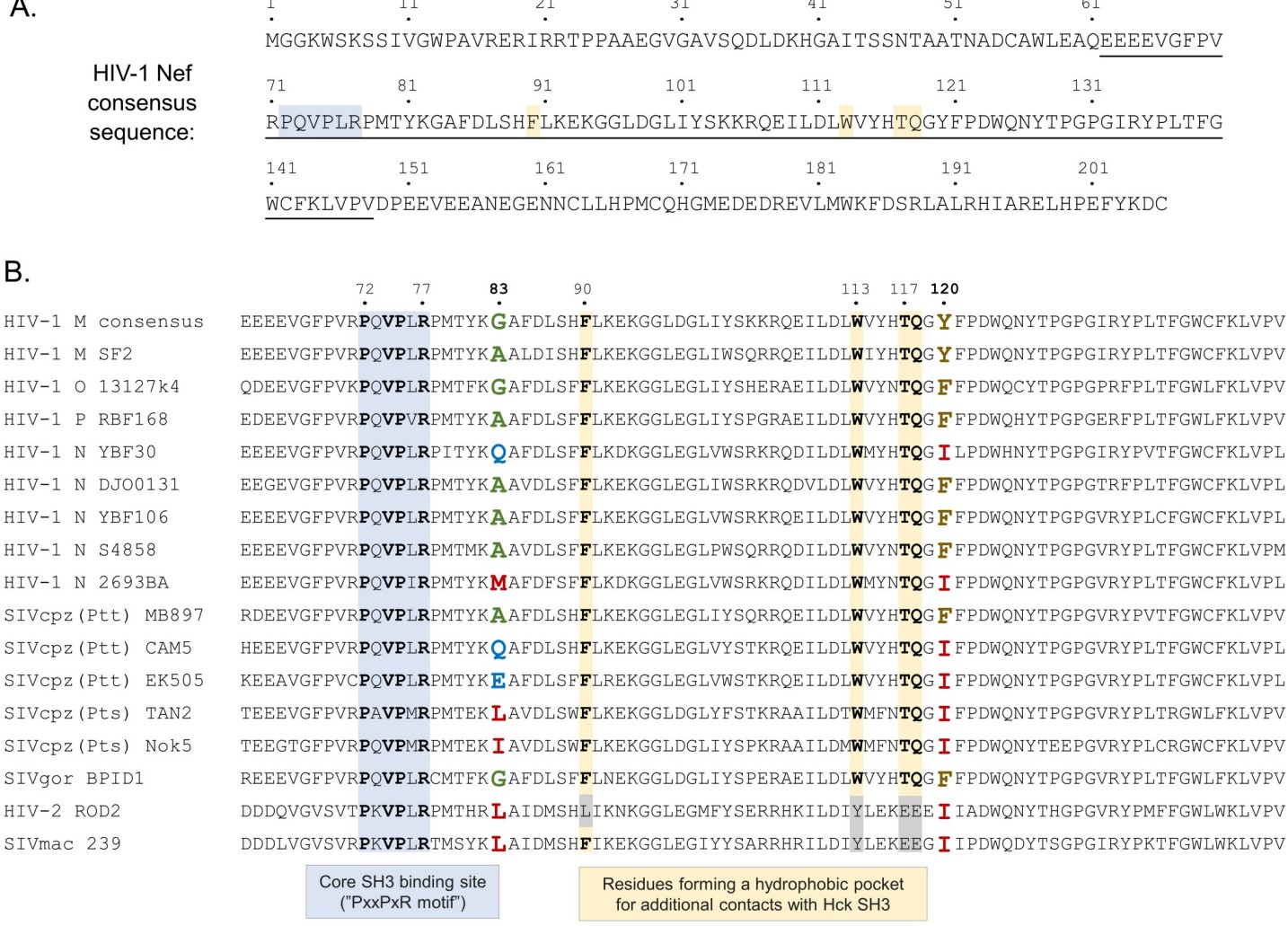

**Fig 2.** HIV-1 M consensus Nef amino acid sequence (**A**) and an alignment of the conserved central region (underlined in A) of selected Nef proteins from different primate lentiviruses (**B**). The key residues of the core SH3 docking site (PxxPxR motif), including R77, are highlighted in light blue. Residues involved in a binding pocket for the Hck SH3 domain RT-loop that are conserved in HIV-1 and HIV-1-like Nef proteins are highlighted in yellow. Residues at positions 83 and 120 forming the R-clamp coordinating the positioning of R77 are shown in bold and colored according to their side chain properties (small in green, hydrophilic in blue, aliphatic or methionine in red, aromatic in brown). The names of HIV/SIV strains from which Nef proteins were included in this study are shown on the left.

carrying the I120 residue 2693BA Nef had an undiminished capacity to activate Hck (Fig 4). In order to understand this finding, we returned to the Nef-SH3 structure 1EFN, and noted the involvement of Nef residue 83 in contacting the critical R-clamp residue 120 (Fig 3). Residue 83 is an amino acid with a tiny (Ala or Gly) or a hydrophilic side-chain (Gln, Ser, Asp, or Glu) in virtually all Nef proteins that have an aromatic (Phe or Tyr) residue at position 120. However, in 2693BA Nef position 83 is occupied by methionine, a residue with a large hydrophobic/aliphatic side-chain. We therefore hypothesized that M83 might compensate for the presence of isoleucine instead of a planar aromatic residue at position 120 (Figs 3D and S4), and that the M83/I120 residue pair of 2693BA Nef might be functionally equivalent to the A83/F120 pair found in the Hck-activating HIV-1 N Nef proteins 02CM-DJO0131 and YBF106.

To test this idea, we swapped residues 83 between the I120-containing 2693BA and YBF30 Nef proteins. In line with our hypothesis, this reversed the differences in the Hck-activating

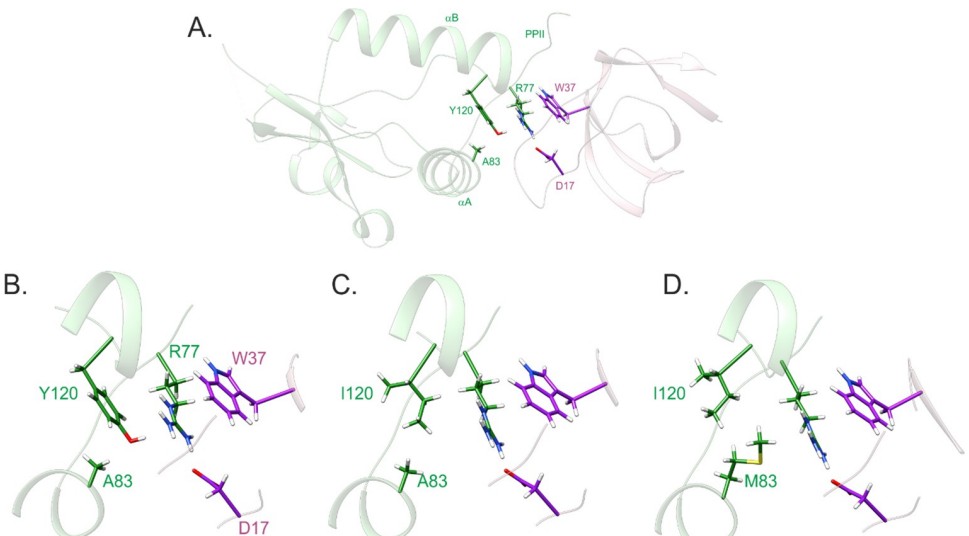

**Fig 3. The arginine clamp. (A)** Structure of the HIV-1 Nef (green)–SH3 (pink) complex (PDB ID 1EFN) highlighting critical residues in the interaction interface. **(B)** Close up view of the R clamp, stabilized by stacked Y-R-W side chains and a salt bridge between R77 (Nef) and D17 (SH3) side chains. **(C)** In the Y120I structure, one side of the R77 guanidinium plane is lacking non-bonded interactions. **(D)** In the A83M/Y120I, structure the long methionine side chain re-establishes non-bonded contacts for R77. The presented mutated complex structures were created by replacing Y120 or Y120 and A83 in UCSF Chimera [45] and minimizing the structures with AMBER [46].

capacity of these HIV-1 N Nefs, such that Nef-2693BA lost its ability to stimulate AP-1 activity, whereas YBF30 Nef became fully competent for this function (Fig 4). Thus, we conclude that Nef residues 83 and 120 appear to function together to form the R-clamp that coordinates Hck-SH3 binding by Nef, and that different combinations of residue pair 83/120 have evolved for building a functional R-clamp in HIV-1 group N Nefs.

Whereas among the Nef proteins of HIV-1, Ile can be found at position 120 only in group N viruses, we noted that several SIVcpz Nefs also contain Ile120 (Table 1). To test if our conclusions regarding the alternative R-clamp designs hold true also for SIVcpz Nef proteins, we examined five of these, three from SIV of the chimpanzee subspecies *Pan troglodytes troglodytes* (P.t.t.), namely MB897 (EF535994), CAM5 (AJ271369), and EK505 (DQ373065), and two from *Pan troglodytes schweinfurthii* (P.t.s.), namely Tan2 (EF394357) and Nok5 (AY536915).

**Table 1. R-clamp residue combinations in the HIV-1-like superfamily of Nef proteins.**

| Category | Residue 83 | Residue 120 | HIV-1 M | HIV-1 O | HIV-1 P | HIV-1 N | SIVcpz (p.t.t.) | SIVcpz (p.t.s.) | SIVgor |
|---|---|---|---|---|---|---|---|---|---|
| | | | n = 5221 | n = 56 | n = 4 | n = 12 | n = 18 | n = 12 | n = 6 |
| I | A,G | F,Y | 96.1% | 100% | 100% | 58.3% | 66.7% | 0% | 100% |
| II | D,E,Q,S | F,Y | 2.6% | 0% | 0% | 0% | 0% | 0% | 0% |
| III | D,E,Q,S | I | 0.5% | 0% | 0% | 25.0% | 33.3% | 0% | 0% |
| IV | I,L,M,V | I | 0.1% | 0% | 0% | 16.7% | 0% | 100%* | 0% |

Altogether, 5329 HIV-1, SIVcpz, and SIVgor Nef sequences were classified into four categories (I–IV) based on the side chain properties of residues 83 and 120 as indicated. Individual HIV-1 M Nef proteins evading this classification could be found, but showed frequencies <0.05%.

*Instead of Ile, some SIVcpz(P.t.s.) proteins contain Val at position 120.

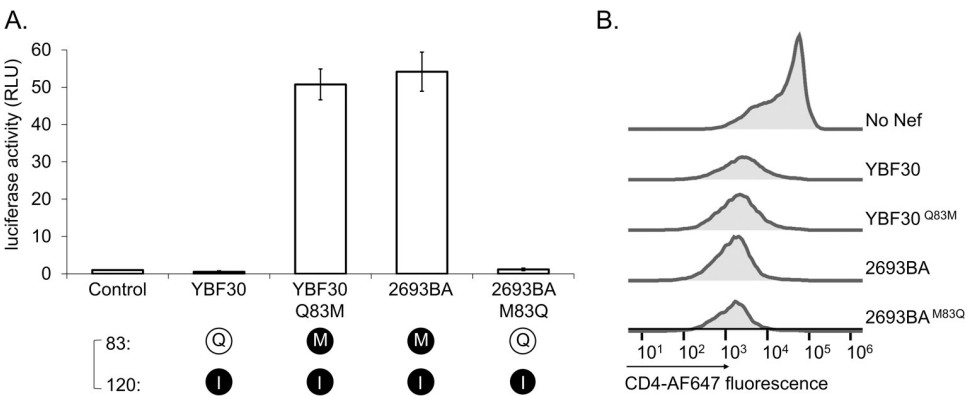

**Fig 4. Role of residue 83/120 pairing in the HIV-1 N Nef R-clamp. (A)** HZ-1 cells were transfected with an AP-1-dependent luciferase reporter alone (Control) or together with wild type (WT) versions of YBF30 or 2693BA Nef, or mutants thereof carrying reciprocal amino acid changes of residue 83 (YBF30 Q83M and 2693BA M83Q). Luciferase activity was measured from cells 4 h post-transfection, and normalized to the corresponding control sample that was set to one. The data shown are derived from three independent experiments, with SE indicated by error bars. The amino acid combinations at positions 83 and 120 are indicated here and in the figures below as single-letter symbols and color-coded according to the residue classification shown in Table 1. Specifically, small (G or A) or hydrophilic residues (D, E, Q, or S) at position 83 are shown as a black 1-letter symbol in a white sphere (here Q83 in YBF30), whereas aliphatic or methionine residues at this position are shown as a white letter in a black sphere (here M83 in 2693BA). At position 120, aromatic residues (F or Y) are shown as a black letter in a white sphere (not found in YBF30 or 2693BA), whereas isoleucine is shown as a white letter in a black sphere (here I120 in YBF30 and 2693BA). **(B)** CD4 downregulation by Nef in stably transduced Jurkat T cells was examined as in Fig 1C. Original dot plots are shown in S2 Fig.

We found that similar to the HIV-1 N YBF30 Nef, some of these SIVcpz Nefs (Cam5 and EK505) failed to activate Hck, whereas others (MB897, Nok5, and Tan2) were functional in this regard although all were able to downregulate CD4 (Fig 5). When the Hck activating capacity was compared with the residue 83/120 composition of these SIVcpz alleles, a perfect agreement was found with the structural R-clamp principles deduced from the results obtained

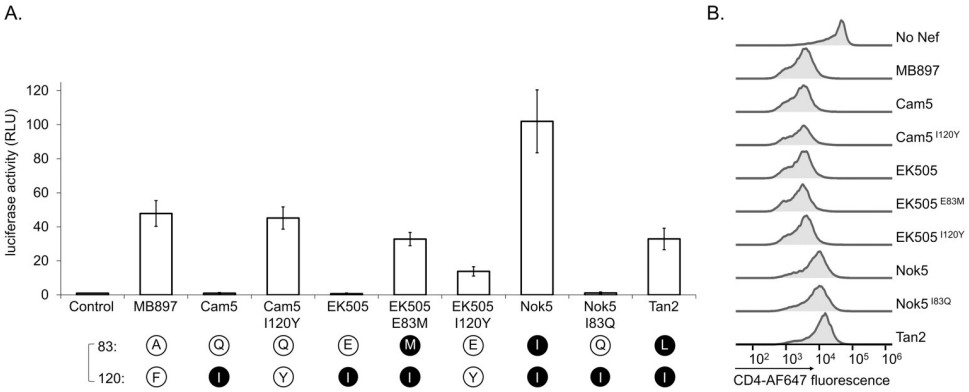

**Fig 5. R-clamp architecture of SIVcpz Nef proteins. (A)** HZ-1 cells were transfected with an AP-1-luciferase reporter alone (Control) or together with wild-type or the indicated mutants of Nef from SIV strains from *Pan troglodytes troglodytes* (MB897, Cam5, and EK505) or *Pan troglodytes schweinfurthii* (Nok5 and Tan2). Luciferase activity in the transfected lysates was analyzed as in Fig 1C. The data shown are derived from three independent experiments, with SE indicated by error bars. The amino acid combinations at positions 83 and 120 are indicated as single-letter symbols and color-coded as explained in Fig 4. **(B)** CD4 downregulation by Nef in stably transduced Jurkat T cells was examined as in Fig 1C. Original dot plots are shown in S3 Fig.

with HIV-1 N Nefs. All SIVcpz Nef proteins that were competent for Hck activation contained a "permissible" [83] + [120] residue combination defined as [tiny/hydrophilic] + [Phe/Tyr] or [aliphatic] + [Ile], whereas both of the two non-activating SIVcpz Nef proteins contained a mixed pattern of amino acid classes at these positions (Figs 2 and 5).

To further prove this concept, we showed that the failure of Cam5 and EK505 Nef to activate Hck could indeed be corrected by introducing a tyrosine residue at position 120 in these proteins, thereby providing them with an R-clamp residue combination 83 [hydrophilic] + 120 [Phe/Tyr]. Moreover, an E83M mutation (resulting in an 83 [aliphatic] + 120 [Ile] residue combination) conferred this activity to EK505 Nef. On the other hand, introducing a hydrophilic residue at position 83 of Nok5 Nef while maintaining its I120 residue (Nok5 I83Q) led to a complete loss of its Hck-activating potential, as predicted by our model on the compatible and non-compatible R-clamp residue combinations.

When the residue 83/120 composition was analyzed in a comprehensive survey of HIV-1/SIVcpz/SIVgor Nef sequences, it was interesting to observe that the different evolutionary lineages of primate lentivirus are associated with distinct R-clamp design strategies. In Table 1, these strategies have been grouped into four categories (I–IV) based on the amino acid side chain properties explained above, and their occurrence has been calculated as a percentage of all Nef sequences found for each lentivirus lineage. Whereas HIV-1 M, HIV-1 O, HIV-1 P, and SIVgor viruses fall almost exclusively into category I, HIV-1 N and SIVcpz viruses show more heterogeneity. Interestingly, however, all SIVcpz Nef proteins from the chimpanzee subspecies P.t.s. fall into category IV, while all SIVcpz(P.t.t.) Nef proteins are found in categories I (67%) or III (33%). HIV-1 N Nef proteins share the R-clamp design patterns with SIVcpz(P.t.t.) and SIVcpz(P.t.s.), and can be found spread between categories I, III, as well as IV. Finally, exclusively HIV-1 M Nef proteins can be found in category II, which includes a small but significant proportion (2.6%) of all the 5221 HIV-1 M Nef protein sequences in our survey.

Of note, the relative distribution of HIV-1 M Nef sequences into categories I to IV is not even between different viral subtypes. Whereas subtype B that constitutes more than half of all available HIV-1 M Nef sequences rarely falls in I120-containing categories III (0.31%) or IV (0.03%), this is much more common among subtype D and G Nef proteins, of which 2.91% and 2.59%, respectively belong to categories III or IV. And although only four of out the total 5221 HIV-1 M Nef sequences belong to category IV, two of these can be found among the 206 subtype D Nef sequences. On the other hand, while category II covers 2.6% of total HIV-1 M Nef sequences, this is the case for as many as 15.6% of subtype G and 7.1% of subtype H, but only 0.5% of subtype A strains.

Based on the experiments presented above it was inferred that Nef proteins in categories I, II, and IV can bind Hck to stimulate its tyrosine kinase activity, whereas category III Nef proteins with a "mixed" R-clamp residues pattern fail to do so. To further substantiate the generality of this conclusion, we introduced several additional individual and combinatorial 83/120 residue changes into the model Nef protein of HIV-1 M SF2 according to the classification presented in Table 1. We found that a Gln, Glu, or Ser residue could indeed all be introduced at position 83 of SF2-Nef without losing its Hck-activating function, as long as an aromatic residue was maintained at position 120 (Fig 6; SF2 A83Q, A83E, and A83S). Thus, all these different category II configurations could support a functional R-clamp when artificially introduced into SF2-Nef.

By contrast, and again in full agreement with our R-clamp model, combination of these mutations with an Y120I change to introduce a category III "mixed" residue 83/120 pattern into SF2 (Q83S/Y120I, A83S/Y120I, and A83E/Y120I) resulted in all three cases in the loss of Hck activation. Finally, the disrupted Hck-activating function of the single residue 120 mutant SF2 Y120I could be fully rescued via an additional introduction of Met, Ile, or Leu at position

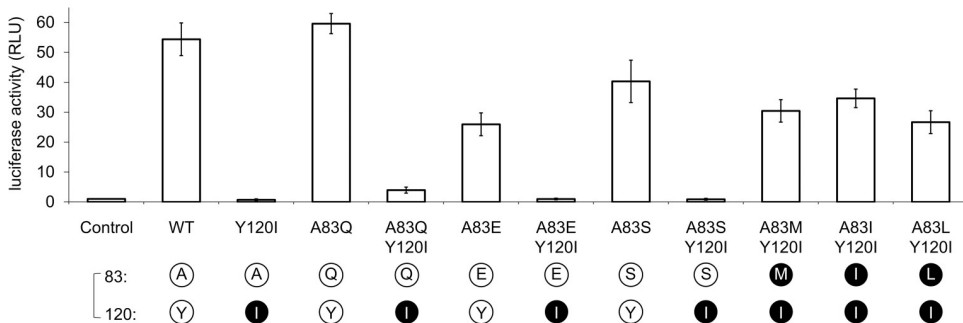

**Fig 6. Comprehensive testing of R-clamp residue pairing rules by mutagenesis of SF2-Nef.** Nef-induced AP-1 activity in cells transfected without Nef (Control) or with the indicated wild-type or mutant versions of SF2 Nef was analyzed as in Fig 1B. The amino acid combinations at positions 83 and 120 of these SF2 Nef variants are indicated as single-letter symbols that are color-coded as explained in Fig 4.

83 to generate artificial category IV-like SF2 Nef double mutants A83M/Y120I ("2693BA-like"), A83I/Y120I ("Nok5-like"), and A83L/Y120I ("Tan2-like").

To confirm that the effects of the Nef R-clamp mutations on the observed cellular changes downstream of Hck activation were indeed due to altered formation of physical Nef-Hck complexes, we examined how wild-type and R-clamp mutated Nef proteins coprecipitated with Hck-p59 from transfected cells. As shown in Fig 7A, association with Hck was observed only for Nef proteins having a functional R-clamp, whereas category III R-clamp Nef proteins were as defective in this regard as the Nef-AxxA mutant carrying a fully disrupted SH3-binding motif.

In addition to its Class II binding motif (PxxPxR) a hydrophobic pocket in the core domain Nef is important for the Hck interaction by making contacts with the RT-loop region of Hck

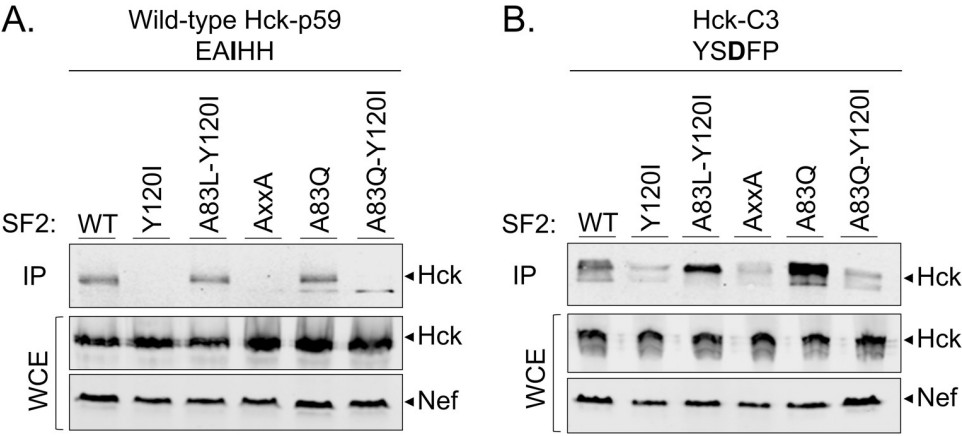

**Fig 7. Role of R-clamp residues 83/120 in intracellular Hck-Nef complex formation.** HEK293 cells were transfected with Myc-tagged wild-type HIV-1 M SF2 Nef (WT) or its mutants including Y120I, A83L-Y120I, P76A-P78A (AxxA), A83Q, and A83Q-Y120I together with biotin acceptor domain-tagged wild-type Hck-p59 (WT) **(A.)** or its SH3 mutant Hck-C3 **(B.)**. The Nef-contacting RT-loop residues centered around the critical SH3 amino acid 13 (Ile in WT and Asp in C3) are shown on top of these figures. Lysates of transfected cells were subjected to anti-Myc immunoprecipitation followed by Western blotting analysis of the immune complexes using labeled streptavidin (top panels). Equal Hck and Nef expression in the total lysates was confirmed by labeled streptavidin (Hck) (middle panels) or an anti-Myc (Nef) antibody (bottom panels).

SH3 [9], in particular the isoleucine residue 13 (universal SH3 numbering according to [18]). Because the R-clamp residues 83 and 120 are part of the compact core of Nef, we wanted to exclude the possibility that mutations of these residues might influence Hck binding by disrupting the stabilizing contacts between Nef and I13 of Hck SH3. To this end, we constructed modified versions of Hck-p59 having artificial SH3 domains containing RT-loops that lack I13 and interact with Nef using diverse non-Hck-like molecular strategies [23,24]. Despite their dissimilar contacts with the hydrophobic pocket of Nef, all of these SH3-modified Hck proteins shared the wild-type Hck-like dependence on a functional R-clamp for interacting with Nef. These data are shown in Fig 7B for the Hck mutant C3 where the native RT-loop residues EA**I**HHE have been replaced with YS**D**FPW (notably containing aspartate, a charged hydrophilic residue at the SH3 position 13 instead of an isoleucine), and similar data for additional two SH3-modified Hck proteins are provided in S5 Fig

To examine the role of the Nef R-clamp in cellular system that is more relevant for HIV biology than HEK293 cells, we tested the capacity of different lentivirally to induce cellular activation in vitro differentiated THP-1-derived human macrophagic cells (Fig 8). This model for macrophage activation depends on SH3 binding by Nef, and is based on monitoring of the activation of the MAPK signaling cascade using phosphorylation of Erk1/2 as the read-out [19]. As evident from two independent experiments involving a set of YBF30 (HIV-1 N) and Nok5 (SIVcpz(P.t.s)) Nef proteins (Fig 8A) or a panel of SF2 Nef variants (Fig 8B), the capacity to activate macrophages and induce their Erk phosphorylation over the steady-state background levels required a functional R-clamp configuration.

Finally, we also studied the role of the R-clamp in another well-established cellular function of Nef, namely enhancement of HIV-1 particle infectivity [25,26]. The underlying mechanism involving counteraction of the SERINC5 restriction factor is not SH3 dependent [27]. However, disruption of the PxxP motif has been associated with a modestly lowered capacity of Nef to increase virion infectivity [28]. When a representative panel of HIV-1 M, HIV-1 N, and SIVcpz(P.t.s) Nef alleles with wild-type or altered residues 83/120 were tested for their ability

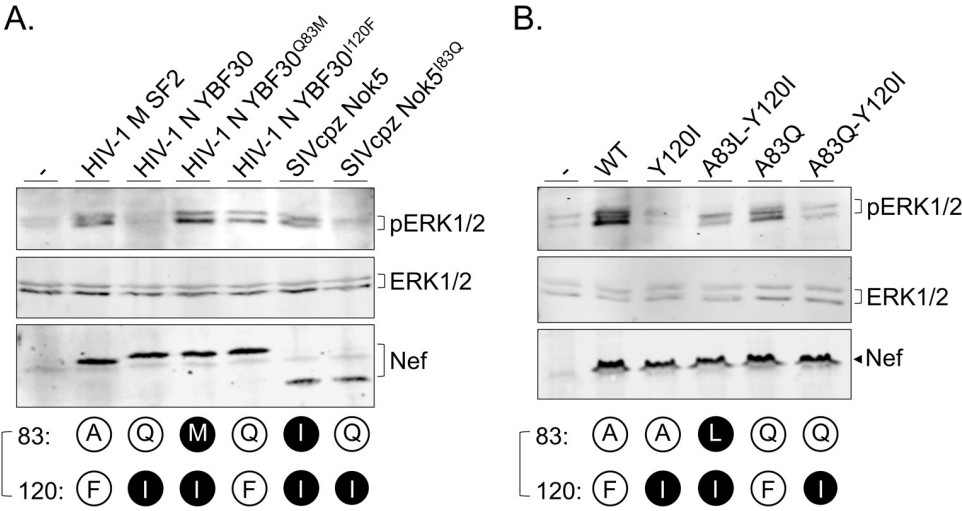

**Fig 8. Role of the R-clamp in activation of in vitro differentiated macrophages by Nef.** The indicated wild-type or mutant Nef proteins from HIV-1 M (panels A. and B.), HIV 1 N (panel A.), or SIVcpz(P.t.s) (panel A.) were expressed via lentiviral transduction in M1-like macrophages differentiated from THP-1 cells, followed by Western blotting analysis to compare the expression of Nef, total ERK1/2, and phosphorylated ERK1/2 in these cells or control macrophages transduced with an empty lentivector (-).

to increase the infectivity of HIV-1 particles (S6 Fig) no apparent differences were observed. This supports the idea that enhancement of HIV-1 infectivity does not depend on SH3 binding by Nef, and confirms our conclusion based on CD4 downregulation that R-clamp inactivating mutations do not universally affect Nef functionality.

Together, our results provide strong further proof to our conclusions on the 83/120 residue combinations that are required for Hck binding and activation, and demonstrate interesting structural and evolutionary plasticity in organizing a set of molecular interactions via an R-clamp principle to enable Nef to tightly bind the SH3 domain of Hck.

## Discussion

Apart from the regulation of trafficking of CD4 and other membrane proteins, the majority of all cellular functions described for Nef depend on its SH3 domain binding capacity, and are lost if the highly conserved consensus SH3 ligand motif PxxPxR of Nef is mutated [7]. Here we show that Nef proteins of HIV-1 and closely related SIVs have evolved a sophisticated molecular mechanism that we have termed the R-clamp in order to coordinate their SH3 binding. This term was coined due to buttressing of the arginine residue of the Nef PxxPxR motif (R77) by the coordinated action of the Nef residues 83 and 120 together with a tryptophan residue (W37) that is conserved in almost all SH3 domains. This places R77 of Nef into close proximity with another highly conserved SH3 residue (D17) to form a salt bridge that stabilizes the Nef/SH3 complex. Thus, docking of proline-rich ligand peptides presented by native proteins, such as Nef, can involve an additional level of structural complexity for tuning of SH3 binding affinity and specificity that is not evident from studies with isolated SH3 binding peptides.

Our studies have focused on the SH3-mediated interaction between Nef and the tyrosine kinase Hck leading to enzymatic activation of Hck, which we have monitored based on induction of downstream signaling events, including paxillin phosphorylation as well as triggering of the Raf/MAPK cascade and subsequent AP-1-regulated gene expression. Nef/Hck interaction is thought to play an important role in HIV infection of cells of the myeloid lineages, such as macrophages, and is an especially tractable study system because of the high affinity of Hck SH3 domain for Nef [19]. This is due to the additional affinity brough into this interaction by tertiary contacts outside of the canonical SH3 ligand binding surface and involving the tip of the RT-loop in Hck SH3 [9,23]. Because of the lack of such RT-loop contacts, SH3 domains of other host cell proteins bind to Nef with a lower affinity resulting in more transient interactions that have remained less well characterized, but appear to include at least Lck [29], Vav [30], PACS-1 [31], Btk, and Itk [32]. Nevertheless, similar to Hck-binding, most of the other SH3 interactions by Nef are also expected to depend on formation of a salt bridge between R77 of Nef and acidic (D or E) SH3 residue 17. Thus, all such interactions, including the SH3-mediated complexes that are relevant for Nef in T lymphocytes depend on a functional R-clamp. However, it cannot be ruled out that despite containing a canonical Class II SH3 binding motif (PxxPxR) Nef (including category III) might also engage in unorthodox interactions with (an) atypical SH3 domain(s) that would not involve R-clamp coordination of R77.

Interestingly, we found that in different lentiviral Nef proteins the R-clamp has been assembled based on alternative designs, which depend on the combination of amino acids at positions 83 and 120. Moreover, the preferred 83/120 residue combinations are dissimilar in different lineages of HIV-1, SIVcpz, and SIVgor (Table 1). While the vast majority of HIV-1 group M, O and P Nefs belong to category I, HIV-1 N Nefs are more heterogenous and fall into categories I or IV (harboring an active R-clamp) or inactive category III (Fig 9). Why would about 25% of all HIV-1 N Nefs harbor an inactive R-clamp, whereas their counterparts from HIV-1 groups M, O, and P do not? Of note, SIVcpz(P.t.t.), the simian precursor of HIV-

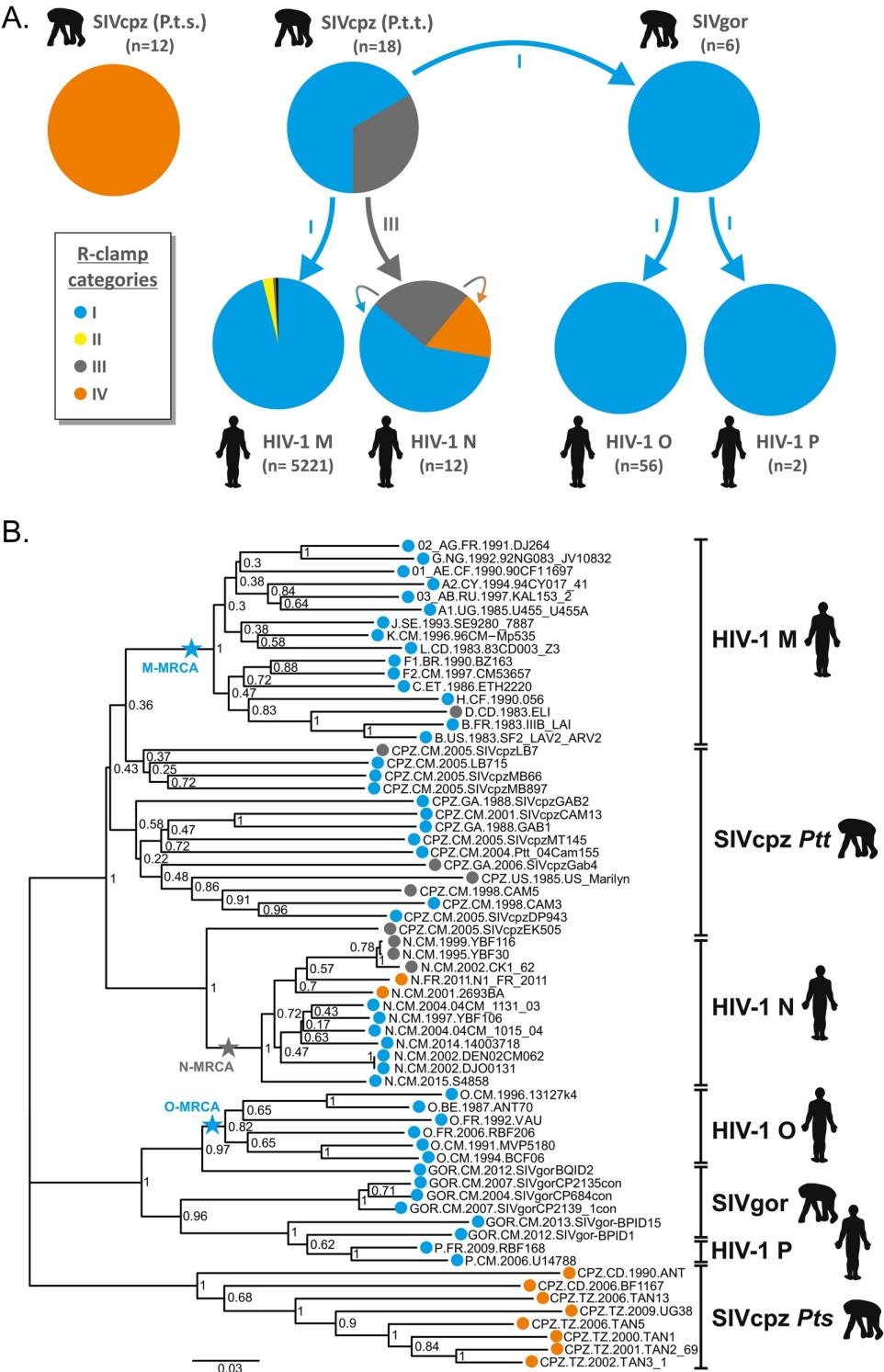

**Fig 9. Putative cross-species transmission and evolution of R-clamp categories.** The pie charts on the left illustrate the relative fraction of each R-clamp category in SIVcpz, SIVgor and HIV-1 groups M, N, O and P. Putative cross-species transmission events are indicated by colored arrows. The lower part of the figure shows a phylogenetic analysis of selected primate lentiviral Nefs is shown. Inferred most recent common ancestors are indicated by stars. In both images, R-clamp categories I, II, III, and IV are shown in blue, yellow, grey and orange, respectively.

1, comprises category I and III Nefs. Based on the phylogenic analysis of primate lentiviral Nef sequences shown in Fig 9B, we propose that the zoonotic transmission that gave rise to HIV-1 group N involved a virus carrying a Nef with a category III R-clamp.

In line with this hypothesis, as indicated in Fig 9B the inferred common ancestor of HIV-1 group N viruses belongs to category III as its Nef protein harbors a combination of Q83 and I120 [33](GenBank accession: KP059120.1). Furthermore, the SIVcpz(P.t.t.) isolate EK505 (DQ373065), a close relative of HIV-1 group N viruses [34] also expresses a category III Nef. Thus, HIV-1 group N viruses may all be the result of a category III virus, and still be adapting and evolving into categories I and IV with functional R-clamps. In this regard it should be noted that group N most likely represents the evolutionarily youngest of all HIV-1 groups [35,36], and be appreciated that the three most recently isolated group N Nefs (JN572926, MF767262, KY498771), including that of N1_FR_2011, the only group N virus isolated outside Cameroon [37], belong to categories I or IV. A single amino acid change of residue 83 (like the EK505-E83M mutant created in this study; see Fig 5) would have been sufficient to move from category III to IV, whereas the path from category III to I would require changes in both residues 83 and 120, and thus follow an evolutionary trajectory via category II. None of the currently available HIV-1 N Nef sequences fall into the R-clamp category II, but given the scarcity (n = 12) of these sequences this does not mean that such HIV-1 N Nefs could still not exist.

In contrast to HIV-1 N, group M viruses almost exclusively fall into category I, suggesting that the zoonotic jump from SIVcpz(P.t.t) to HIV-1 M involved a virus with a category I Nef R-clamp. Indeed, the inferred most recent common ancestor of HIV-1 M Nefs (KP059118.1; KP059119.1; [33] belongs to category I. This is also true for the oldest known HIV-1 M Nef (M15896) isolated in 1976 [38]. Finally, the exclusive use of category I R-clamp by Nef proteins of SIVgor, HIV-1 O, and HIV-1 P would suggest that an R-clamp category I virus might have originally been transmitted from chimpanzee to gorilla, and subsequently in two independent cross-species jumps entered humans, giving rise to HIV-1 O and HIV-1 P.

It is of interest to note that a subtype D virus of the HIV-1 M group has been previously shown to encode a Nef protein unable to activate Hck, and that this deficiency was mapped to the unusual presence of an isoleucine residue at the Nef position 120 [39]. This study suggested that Y120 would contribute to the hydrophobic pocket of Nef that accommodates the Hck SH3 domain RT-loop, whereas I120 could not serve this function. However, these molecular contacts inferred based on our (NL4-3) Nef/SH3 X-ray structure (1EFN; [9]}) would be expected to only minor rather than critical for the Nef-Hck interaction, and a recent SH3 complex structure involving HIV-1 SF2 Nef [40] does not support such a role for Y120 at all. Moreover, our current data clearly show that category IV Nef proteins containing I120 do bind and activate Hck when paired with an appropriately matched R-clamp residue at the position 83. Thus, while the findings of Choi and Smithgall on Nef from HIV-1 ELI [39] agree with our data, they need to be interpreted in light of the R-clamp concept described here.

While the evolutionary scenarios discussed above may help to explain the occurrence of inactive category III Nefs in HIV-1 group N, it still remains unclear why about one third of all SIVcpz(Pt.t.) Nefs harbor a category III R-clamp. Furthermore, the reasons for the unique predominance of category IV Nefs in SIVcpz(P.t.s.) remain to be determined. Apart from the initial founder viruses that gave rise to the respective lentiviral lineages and species, the optimal R-clamp composition and the rate of R-clamp sequence evolution are probably determined by a complex balance between host-specific factors and sequence variation elsewhere in these viruses. To gain further insights, it would be interesting to examine a larger number of HIV-1 N Nef sequences over time and from different tissues from the same individual to understand whether the R-clamp pattern is relatively fixed or subject to rapid evolution and quasispecies variation. In the latter case, because of the highly multifunctional nature of the Nef protein, a non-functional

R-clamp might provide a selective advantage in certain anatomic locations or special circumstances that might occur during or immediately after the zoonotic jump from chimpanzee to man. On the other hand, as already noted, despite the virtually universal conservation of an intact Class II SH3 binding motif PxØPxR (where Ø is a hydrophobic residue) in all Nef proteins (including R-clamp category III), it could be speculated that while unable to interact with canonical SH3 domains, R-clamp category III Nef proteins might instead show unusual specificity for some atypical SH3 proteins, which could provide them with alternative cellular functions.

In any case, R-clamp sequence variation provides a fascinating example of evolutionary plasticity of a protein interaction interface and our findings suggest that the selection pressures that have shaped Nef during primate lentiviral evolution are different depending on the HIV/SIV lineage and their host species. The dynamic capacity of Nef for altering the molecular strategy of recognizing its key host cell interaction partners, such as SH3 domains, is also important to keep in mind in attempts to develop novel HIV eradication therapies that might target the immune evasion function of Nef.

## Materials and methods

### Reagents and cell lines

Mouse anti-Myc (sc-40) and rabbit anti-paxillin (sc-5574) were from Santa Cruz Biotechnology. Mouse anti-GAPDH (607902) was from Biolegend). Mouse anti-pY31 paxillin was from BD Biosciences. Rabbit anti phospho-p44/42 MAPK (ERK1/2) (9101) and rabbit anti- p44/42 MAPK (ERK1/2) (9102) antibodies were from Cell Signaling Technology. Alexa Fluor 647 conjugated anti-human CD4 antibody was from SouthernBiotech. IRDye680CW goat anti-mouse IgG, IRDye800CW Streptavidin, and IRDye800CW goat anti-rabbit IgG were from LI-COR Biotechnology. HEK293, HEK293T, THP-1, and Jurkat cells were obtained from ATCC. TZM-bl cells are a HeLa-derived reporter cell line and were obtained through the NIH AIDS Reagent Program, Division of AIDS, NIAID, NIH, from John C. Kappes, Xiaoyun Wu, and Tranzyme Inc. [41]. The derivation and characterization of the HEK293-based Hck-expressing HZ-1 cells have been described elsewhere [19]. HEK239T, HEK293 and HZ1 cells were grown in high-glucose Dulbecco's modified Eagle's medium (DMEM; Sigma) supplemented with 10% fetal bovine serum (FBS), 0.05 mg/ml penicillin, and 0.05mg/ml streptomycin. HEK293T cells were transfected in 6 well plates using a standard calcium phosphate method. Lentiviral transduction was employed to generate Jurkat cells stably expressing human CD4. Briefly, HEK293 cells were co-transfected with 2.5 μg pDelta8.9, 1.5 μg VSV-G and 3 μg pWPI-puro plasmid containing human CD4 cDNA in Opti-MEM medium with 12 μg PEI. After 5 hours, medium was refreshed with cell culture media. Supernatant was collected 48 hours post-transfection, filtered and used to infect Jurkat cells. Infected cells were selected with 6 μg/ml puromycin for 2 days. Single cell clones of CD4 transduced Jurkat cells were isolated by cellenONE X1 system. THP-1, THP-1 derived macrophages, and Jurkat cells were maintained in RPMI-1640 medium (Sigma) supplemented with 10% FBS, 0.05 mg/ml penicillin, and 0.05 mg/ml streptomycin.

### Plasmids

*Nef* alleles from HIV-1 M SF2 (P03407), HIV-1 N 2693BA (GQ925928), HIV-1 N S4858 (KY498771), HIV-1 N YBF30 (AJ006022), HIV-1 N YBF106 (AJ271370), HIV-1 N DJO0131 (AY532635), HIV-1 O 13127k4 (AY536904), HIV-1 P RBF168 (GU111555), SIVcpz(P.t.t) CAM5 (AJ271369), SIVcpz(P.t.t) EK505 (DQ373065), SIVcpz(P.t.t) MB897 (EF535994), SIVcpz(P.t.s) ch-Nok5 (AY536915) and SIVcpz(P.t.s) Tan2 (EF394357) were cloned into the expression vector pEBB containing a C-terminal Myc-tag. Human Hck p59 cDNA (isoform b/

NP_001165604) was cloned into the pEBB vector together with a 123 aa biotin acceptor domain fused to their C-termini [21]. Hck variants A1, B6, and C3 [25] were created to the same vector background. The Nef mutants (HIV-1 M SF2 Y120F, Y120I, A83E, A83E/Y120I, A83I, A83I/Y120I, A83L, A83L/Y120I, A83M, A83M/Y120I, A83Q, A83Q/Y120I, A83S, A83S/Y120I; HIV-1 N YBF30 Q83M, YBF30 I120F, 2693BA M83Q; SIVcpz(P.t.t) Cam5 I120Y, EK505 E83M, EK505 I120Y and SIVcpz (P.t.s) ch-Nok5 I83Q, ch-Nok5 I120F) were generated in the same vector backbone using standard PCR-assisted mutagenesis. All of the Nef variants mentioned above were cloned into the pWPI-GFP vector (Addgene # 12254) for lentiviral transduction of stably CD4-expressing Jurkat cells and THP-1-derived macrophages.

## Immunoblots

Cells were collected and lysed on ice for 10 minutes in lysis buffer (150 mM NaCl, 50 mM Tris-HCl [pH 7.4], 1% NP-40) with protease and phosphatase inhibitors (Thermo Fisher Scientific). Cell lysates were centrifuged at 16,000 x g at 4°C for 5 min. Proteins from cell extracts were analyzed by standard SDS gel electrophoresis and Western blotting using IRDye-labeled detection reagents detailed above.

## AP-1 luciferase reporter assay

HZ1 cells were transfected using TransIT-2020 reagent (Mirus) with 50 ng of Nef expression vector together with 50 ng of AP-1 pfLUC reporter plasmid [42] driving the AP-1 inducible expression of firefly luciferase, plus 50 ng of the plasmid pRL-TK (Promega) expressing low and constitutive levels of *Renilla* luciferase. Cells were collected and lysed with lysis buffer (Promega) on ice. A dual-luciferase reporter assay system from Promega was utilized to determine luciferase activities following the manufacturer's protocol using Berthold Sirius single-tube luminometer detection.

## Lentiviral transduction of THP-1 derived macrophages

$1 \times 10^8$ THP-1 cells were infected with lentiviral vectors containing various *nef* alleles. 3 days post-infection, GFP positive THP-1 cells were sorted by fluorescence-activated cell sorting (FACS). Subsequently, $1 \times 10^7$ sorted THP-1 cells were seeded into each well in a 6-well plate and treated with 10 ng/ml of PMA for 2 days. Adherent cells (M0 macrophages) were further cultured for 2 days in the presence of 10 ng/ml PMA and 10 ng/mL of granulocyte-macrophage colony-stimulating factor (GM-CSF) (ThermoFisher) to differentiate them towards an M1 macrophage phenotype.

## Flow cytometry analysis of CD4 downregulation

Nef-expressing HIV-1-like pseudoviruses were produced as described previously [19]. Jurkat cells stably expressing CD4 were infected with such lentiviral vectors containing various *nef* alleles. 3 days post-infection the cells were collected and washed twice with PBS (pH 7.4), followed by fixing with 1% Paraformaldehyde (Sigma) at room temperature for 20 min. Cells were washed twice with PBS containing 2% FBS (FACS buffer) and stained with Alexa Fluor 647 conjugated anti human CD4 antibody at room temperature for 40 min. After staining the cells were washed twice with FACS buffer and re-suspended in PBS. The CD4 cell surface expression was analyzed using a BD Accuri C6 flow cytometer (BD Biosciences) and data analysis was performed using FlowJo software (version 10.4, Ashland OR: Becton, Dickinson and Company).

## Virion infectivity

To determine the effects of Nef on virion infectivity, HEK293T cells were co-transfected with pEBB expression plasmids for different Nefs or an empty vector control and an HIV-1 reporter virus lacking functional *nef* and *vpu* genes (HIV-1 NL4-3 Δ*nef* Δ*vpu* IRES eGFP). Two days post transfection, cell culture supernatants were harvested. To quantify infectious virus yield, 6,000 TZM-bl cells were seeded in 96-well plates and infected with the cell culture supernatant of transfected HEK293T cells in triplicate on the following day. 3 days post infection, β-galactosidase reporter gene expression was determined using the GalScreen kit (Applied Bioscience) according to the manufacturer's instructions. In parallel, p24 concentration was determined using a home-made ELISA. Relative virion infectivity was subsequently calculated by normalizing infectious virus yield to the amount of p24.

## Phylogenetic analyses

A maximum-likelihood phylogenetic tree was constructed as described previously [43]. Briefly, the 65 nucleotide sequences of the *nef* gene included in the analysis were aligned using MUSCLE and the phylogenetic tree was constructed using MEGA7 [44]. Most recent common ancestors (MRCA) of Nef proteins of HIV-1 groups M, O, and N had been inferred in a previous study [33].

## Supporting information

**S1 Fig. Contour plots illustrating cell surface CD4 expression and HIV-1 Nef transduction (GFP) levels.** Shown are the raw data for the histograms in Fig 1C.
(PDF)

**S2 Fig. Contour plots illustrating cell surface CD4 expression and HIV-1 Nef transduction (GFP) levels.** Shown are the raw data for the histograms in Fig 4B.
(PDF)

**S3 Fig. Contour plots illustrating cell surface CD4 expression and SIVcpz Nef transduction (GFP) levels.** Shown are the raw data for the histograms in Fig 5B.
(PDF)

**S4 Fig. One-hundred ns all-atom molecular dynamics simulations show that stabilization of R77 side chain provided by the R-clamp in the wild-type complex could be reproduced by Met, Ile and Trp in the A83M/Y120I double mutant.** Analysis of the MD trajectories show that in the wild type complex the stacked Tyr-Arg-Trp π-cation-π interaction remains stable, as interpreted from Arg to Tyr and Arg to Trp side chain distances, which show little fluctuations around their average values, 4.2 Å (**A**) and 3.5 Å (**B**). In (**C**) is shown a snapshot from the wild type complex simulation in which these distances are close to their average values. In the double mutant the Arg to Trp distance is on the average about 0.5 Å longer than that in the wild type complex (**E**), but also remains stable around its average value, 4.0 Å. If 6 Å is taken as the maximum distance for a cation-π interaction [47] the observed distances are well within the limit. Likewise, hydrophobic contacts to and between Met and Ile on the other side of the Arg plane remain relatively stable. Non-bonded contacts are likely to be bolstered by Met sulfur [48]. Ile and Arg sidechains are on average 4.4 Å apart (**D**). Met to Arg (**F**) and Ile to Met (**G**) distances show more variation, but are for the majority of time close to about 4.6 and 3.9 Å. In (**H**) is shown a snapshot from the double mutant complex simulation in which these distances are close to their average values. A salt bridge between Arg and Asp side chains is present in 96% (WT) and 93% (double mutant) of the simulation frames. MD

simulations in explicit solvent were performed with AMBER 20 [24] using the ff14SB force field. The WT and A83M/Y120I complexes were placed in a cubic box with a minimum solute-box distance of 10 Å, and solvated with TIP3P water molecules. Six sodium ions were added to neutralize the system. After minimization, heating and equilibration of the system, the production 100-ns MD simulations were performed with periodic boundary conditions at 300 K. The temperature was maintained by using the Langevin thermostat, whereas the pressure was kept at 1 bar using the Berendsen barostat [49]. The time step was set to 2 fs. Long-range electrostatic interactions were treated using the Particle Mesh Ewald method [50] with a cut-off of 10 Å. Bond lengths involving hydrogen atoms were constrained by SHAKE [51]. Analyses of the trajectories were carried out with CPPTRAJ [52].
(PDF)

**S5 Fig. Role of R-clamp residues 83/120 in intracellular Hck-Nef complex formation.** HEK293 cells were transfected with Myc-tagged wild-type HIV-1 M SF2 Nef (WT) or its mutants including Y120I, A83L-Y120I, P76A+P78A (AxxA), A83Q, and A83Q-Y120I) together with biotin acceptor domain-tagged SH3 mutated Hck-p59 variants Hck-A1 (**A.**) or Hck-B6 (**B.**). Similarly, YBF30 or its R-clamp gain-of-function mutant Q83M were co-expressed with biotin acceptor domain-tagged wild-type Hck-p59 or its SH3 mutant Hck-B6. (**C.**). The Nef-contacting RT-loop residues centered around the critical Hck SH3 amino acid 13 (indicated in bold) are shown on top of these Figs. Lysates of the transfected cells were subjected to anti-Myc immunoprecipitation followed by Western blotting analysis of the immune complexes using labeled streptavidin (top panels). Equal Hck and Nef expression in the total lysates was confirmed by labeled streptavidin (Hck) (middle panels) or an anti-Myc (Nef) antibody (bottom panels).
(PDF)

**S6 Fig. Role of the R-clamp in Nef-mediated enhancement of virion infectivity. (A)** HEK293T cells were co-transfected with pEBB expression plasmids for the indicated Nefs and an HIV-1 reporter virus lacking functional nef and vpu genes (HIV-1 NL4-3 Δnef Δvpu IRES eGFP). Two days post transfection cell culture supernatants were harvested. Infectious virus yield was determined by infecting TZM-bl reporter cells and normalized to the amount of p24 (as determined by ELISA) to calculate virion infectivity. Mean values +/- SEM of four independent experiments are shown. **(B)** Two days post transfection cells were harvested for Western Blot analysis. Nef was detected via an anti-myc tag antibody. GAPDH served as loading control.
(PDF)

## Acknowledgments

We thank Ms. Virpi Syvälahti for expert technical assistance.

## Author Contributions

**Conceptualization:** Zhe Zhao, Riku Fagerlund, Frank Kirchhoff, Perttu Permi, Daniel Sauter, Kalle Saksela.

**Formal analysis:** Kalle Saksela.

**Funding acquisition:** Perttu Permi, Daniel Sauter, Kalle Saksela.

**Investigation:** Zhe Zhao, Riku Fagerlund, Helena Tossavainen, Kristina Hopfensperger, Rishikesh Lotke, Smitha Srinivasachar Badarinarayan, Frank Kirchhoff, Perttu Permi, Kei Sato, Daniel Sauter, Kalle Saksela.

**Methodology:** Zhe Zhao, Riku Fagerlund, Helena Tossavainen, Kristina Hopfensperger, Rishikesh Lotke, Smitha Srinivasachar Badarinarayan, Perttu Permi, Kei Sato, Daniel Sauter, Kalle Saksela.

**Project administration:** Kalle Saksela.

**Resources:** Perttu Permi, Kalle Saksela.

**Supervision:** Perttu Permi, Daniel Sauter, Kalle Saksela.

**Writing – original draft:** Zhe Zhao, Riku Fagerlund, Helena Tossavainen, Frank Kirchhoff, Perttu Permi, Daniel Sauter, Kalle Saksela.

**Writing – review & editing:** Zhe Zhao, Riku Fagerlund, Helena Tossavainen, Kristina Hopfensperger, Rishikesh Lotke, Smitha Srinivasachar Badarinarayan, Frank Kirchhoff, Perttu Permi, Kei Sato, Daniel Sauter, Kalle Saksela.

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
