## [Decision Letter · Decision Letter 0]

3 Jul 2021

Dear Dr. Saksela,

Thank you very much for submitting your manuscript "Evolutionary plasticity of SH3 domain binding by Nef proteins of the HIV 1/SIVcpz lentiviral lineage" for consideration at PLOS Pathogens. As with all papers reviewed by the journal, your manuscript was reviewed by members of the editorial board and by several independent reviewers. In light of the reviews (below this email), we would like to invite the resubmission of a significantly-revised version that takes into account the reviewers' comments.

We cannot make any decision about publication until we have seen the revised manuscript and your response to the reviewers' comments. Your revised manuscript is also likely to be sent to reviewers for further evaluation.

Sincerely,

Raul Andino

Section Editor

PLOS Pathogens

Kasturi Haldar

Editor-in-Chief

PLOS Pathogens

orcid.org/0000-0001-5065-158X

Michael Malim

Editor-in-Chief

PLOS Pathogens

orcid.org/0000-0002-7699-2064

Reviewer's Responses to Questions

**Part I - Summary**

Reviewer #1: Zhao and colleagues investigate the interaction between the accessory HIV-1 Nef protein and human SH3 domains. In particular the authors focus on an ionic interaction between an arginine from Nef that is part of its class II proline rich motif PXXPXR, and an aspartic acid in SH3 domains present in their so-called RT loop. The existence of this type of interaction has been known in the Nef–SH3 complex since 1996, and is also a conserved feature of other SH3–ligand interactions. However, Zhao et al. extend this knowledge by showing that the arginine in Nef is structurally supported by what they call an ‘R-clamp’ architecture. They show that the R-clamp structure is preserved, however the residues involved in this mechanism segregate with different HIV and SIV groups. They find that particular ‘R-clamp’ residue combinations abrogate activation of Hck. The atypically strong and selective association that Nef engages with SH3 domains is important for the pathogenicity of immunodeficiency viruses, and hence a better understanding of the molecular and evolutionary basis of how this interaction evolved can be an important factor for understanding HIV biology and directing therapy.

I do think that it is interesting and novel hat Nef’s capacity to promote paxillin phosphorylation by Hck depends (also) on a combination of two Nef residues (in position 83 and 120); and that there are Nef isolates found where this combination is such that Nef cannot longer activate Hck. However based on the data shown, I’m not convinced about their explanation for this observation, as I will explain in the following:

The correlation between physicochemical characteristics of side chains from residues that fill the same 3D space in a protein structure is of course well known (it is used, for example to establish ‘contact maps’ for proteins). It is also well known that mutations that cause gaps in a protein structure, or in a protein complex, negatively affect the stability of the protein or protein complex. Hence the loss-of-function observed by combining two insufficiently large side chains from positions 83 and 120 are expected, as they would leave a gap in the structure. However, having a combination of 83=Q/E and 120=I should fill the available space just the same, and would also allow to support (or ‘clamp’ as the authors say) R77 with a polar interaction. Interestingly, in a putative Hck SH3 complex, Q/E83 would come close to the hydrophobic Hck Ile95, which might explain that this is an unfavourable combination for Hck SH3. When introducing a larger F/Y in position 120, then the long side chain of 83 needs to bend away from Ile96 to avoid clashes, and this might result in a more favourable hydrophobic surface for Hck I95 binding as Q83 only presents parts of its aliphatic side chain region.

So my major concern is that the observations with regard to Hck binding (or their proxy of paxillin phosphorylation) are not explained by the stabilisation (clamping) of R77, but by the attractive or repulsive interactions between Hck Ile95 and Nef 83. It has been shown that some SIV Nef alleles bind weaker to Hck SH3 and stronger to SH3 from Src kinases (Src, Fyn) that have a charged/polar residue in the position of Hck Ile95 (Collette, JBC 2000). Hence, would it be possible that the ‘category III’ Nef evolved to support binding of non-Hck SH3 domains--and that the absence of Hck SH3 binding is not an issue of lacking stabilisation of R77? I think that this alternative hypothesis needs to be addressed/ruled out before the clamping mechanism, and its proposed evolutionary repercussions and conclusions, can be accepted.

E.g. 83/120 combinations that the authors termed inactive might actually present a gain of function towards another (SH3) target. This possibility (which they mention on page 26) would majorly affect their evolutionary discussion.

Reviewer #2: SIV/HIV Nef are multifunctional, highly conserved viral accessory proteins necessary for viral spread and pathogenicity in vivo. Nef exerts various activities in infected cells. Many of them depend on Nef capacity to bind the SH3 domain of cellular proteins via its polyproline (PxxP) domain. The interaction between Nef and the SH3 domain of Hck is well characterized. Hck is expressed in the myeloid lineage, and its activation initiates a cascade of events leading toPaxillin phosphorylation and AP1 activation.

In this study, Zhao and colleagues characterized the molecular details of the PxxP/SH3 interaction. They identified a molecular mechanism that they termed “R-clamp” that regulates Nef/SH3 binding.

Starting from identifying a Nef allele unable to induce Paxillin phosphorylation despite an intact PxxP domain, they identified the residues 83 and 120 in Nef as crucial to stabilize the Nef/Hck interaction. The authors then analyzed various Nef alleles from HIV-1 M, N, O, or P groups and SIV. They identified four “categories of R-clamps” defined by different amino acid combinations and associated with functional or “unfunctional” PxxP. Results were obtained by overexpressing Nef alleles in HZ-1 (HEK293-based Hck-expressing cells) and measuring either Paxillin phosphorylation or AP1 (luciferase-based reporter) activation as a readout of Hck activation by Nef.

Reviewer #3: In this work, the authors discover a new mechanism that assists in lentiviral Ned protein interaction with the host protein SH3 domain.

Many viral proteins evolved short linear motifs that mimic host endogenous motifs, to support in their interactions with host proteins (through domain-motif interactions).

The specificity and affinity of these motifs to host domains are an outstanding question in the field of host-virus interactions since these motifs are usually small and form only few and transient interactions with the host domain. In here, the authors find a new mechanism that they term "buttressing", or arginine R-clamp, where additional interactions are formed, in addition to the ones created by the motif itself, to support the main module of interaction.

I find this idea interesting and novel, and with important consequences to our understanding of viral protein evolution. However, I do have a few reservations about the experimental design.

**Part II – Major Issues: Key Experiments Required for Acceptance**

Reviewer #1: The minimum required would be to do in vitro binding tests of selected Nef representatives (at least category I and III, ideally also from the other categories) with SH3 domains from Hck, Src and Lck, for example (SH3 RT loop residues are Ile, Arg, Ser). ITC, MST, fluorescence anisotropy or other quantitative methods could be used.

Such a complementary in vitro binding assay would also be reassuring, given that the authors used “paxillin phosphorylation” as a proxy for “Nef binding to Hck SH3 domains” which might not be equivalent. E.g. how to rule out that other Src kinases are bound and activated and then phosphorylate paxillin (Src?).

I would also suggest to analyse the protein stability of selected Nef mutants using DSF, DSLS or CD.

Reviewer #2: The manuscript is well written. Results are convincing but preliminary. They would need further confirmation and mechanistic analysis in more relevant cell types.

1. A more relevant cell type should be used to test R-clamp function, such as differentiated THP-1 or primary macrophages.

2. The authors show as a control that the mutations do not affect CD4 downregulation. This is not surprising since the polyproline domain of Nef is not required for this function. Other functions dependent on a functional PxxP domain should be tested, such as MHC-I downmodulation and cell activation.

3. What is the impact of mutations in the R-clamp on viral infectivity?

4. A more detailed phylogenetic analysis of R-clamp evolution in various HIV/SIV groups could be helpful to understand how the authors see R-clamp selection and evolution.

Reviewer #3: For studying the interactions between the host SH3 domain and the viral protein, the authors used an assay where they probed the inferred outcome of the successful interaction – by looking for phosphorylation of paxillin, using Western blotting.

While this assay is convincing it should be supported by a more direct evidence of the formation, or absence, of the interaction, such as using a tagged protein of one of the interactors followed by a pull-down approach to test whether the second interacting partner is indeed bound or not. It seems essential for a work whose focus is on finding a new interaction mechanism to have a more biochemical approach to verify these interactions.

**Part III – Minor Issues: Editorial and Data Presentation Modifications**

Reviewer #1: I found the ‘clamp’ term to be misleading. A clamp implies that an entity is held in place by forces from two opposite sides. However what the authors describe is more a one-sided support of R77. The 3D fixing of R77 is also the result of the Arg side chain with the backbone carbonyl and side chain of Q118. But R77 is not ‘clamped’ by positions 83 and 120.

Figure 1A: the expression levels of those sequences that do not lead to paxillin phosphorylation appear to be the lowest. If so, then this should be noted and discussed.

P14: “the Y120I mutation recapitulated the failure of YBF30 Nef to stimulate paxillin phosphorylation and AP-1 activity, without compromising protein stability or CD4 downregulation (Figure 1).” Here the authors do not measure protein stability, but only CD4 downregulation. Protein stability should be measured in vitro, using DSF or DSLS or equivalent methods.

Table 1: I would put ‘residue 83’ in column 2 and ‘residue 120’ in column 3, to preserve the sequence order.

P21: “this places R77 of Nef into close proximity with another highly conserved SH3 residue (D17) to form a salt bridge that stabilizes the Nef/SH3 complex.” I would suggest to use Hck numbering here, rather than a more arbitrary ‘SH3 numbering’.

P22: “Why would about 25% of all HIV-1 N Nefs harbor an inactive R-clamp, whereas their counterparts from HIV-1 groups M, O, and P do not?” As long as the category III has not been shown to not bind to any SH3 domain in vitro, I don’t think that these Nefs can be catalogued as ‘inactive’.

P24: in the second paragraph, two or three difference findings and probably manuscripts are all referenced as [36]. Is it possible that the first and 3rd ref 36 are actually https://doi.org/10.1016/j.jmb.2004.09.015?

Reviewer #2: (No Response)

Reviewer #3: - The authors focus on one SH3-containing protein - Hck, and only briefly mentioned other proteins that the virus interacts with through this motif. Can a more detailed discussion regarding why this mechanism of arginine R-clamp is not needed in those proteins be added, at least in Discussion (if not in additional experiments)?

- In Introduction:

I suggest changing “pushing” to “shifting”:

Nef tightly binds to the SH3 domain of Hck, thereby pushing it from an intramolecular autoinhibitory state into a catalytically active conformation [9, 10].

PLOS authors have the option to publish the peer review history of their article (what does this mean?). If published, this will include your full peer review and any attached files.

Reviewer #1: **Yes: **Stefan T Arold

Reviewer #2: No

Reviewer #3: No
---

## [Decision Letter · Decision Letter 1]

28 Oct 2021

Dear Dr. Saksela,

We are pleased to inform you that your manuscript 'Evolutionary plasticity of SH3 domain binding by Nef proteins of the HIV 1/SIVcpz lentiviral lineage' has been provisionally accepted for publication in PLOS Pathogens.

Best regards,

Raul Andino

Section Editor

PLOS Pathogens

Kasturi Haldar

Editor-in-Chief

PLOS Pathogens

orcid.org/0000-0001-5065-158X

Michael Malim

Editor-in-Chief

PLOS Pathogens

orcid.org/0000-0002-7699-2064

Reviewer Comments (if any, and for reference):

Reviewer's Responses to Questions

**Part I - Summary**

Reviewer #1: The authors have addressed or at least discussed all points I raised. Although they did not always use the in vitro experiments I suggested, I consider the experiments performed, together with the text amendments made, sufficient.

Reviewer #2: THe authors have adressed my concerns

Reviewer #3: The authors have addressed my main concern regarding a direct evidence for interaction, as well as the minor comments I had.

**Part II – Major Issues: Key Experiments Required for Acceptance**

Reviewer #1: The authors have addressed or at least discussed all points I raised. Although they did not always use the in vitro experiments I suggested, I consider the experiments performed, together with the text amendments made, sufficient.

Reviewer #2: (No Response)

Reviewer #3: (No Response)

**Part III – Minor Issues: Editorial and Data Presentation Modifications**

Reviewer #1: The authors have addressed or at least discussed all points I raised.

Reviewer #2: (No Response)

Reviewer #3: (No Response)

PLOS authors have the option to publish the peer review history of their article (what does this mean?). If published, this will include your full peer review and any attached files.

Reviewer #1: **Yes: **Stefan T Arold

Reviewer #2: No

Reviewer #3: **Yes: **Tzachi Hagai

---

## [Editor Report · Acceptance letter]

9 Nov 2021

Dear Dr. Saksela,

We are delighted to inform you that your manuscript, "Evolutionary plasticity of SH3 domain binding by Nef proteins of the HIV 1/SIVcpz lentiviral lineage," has been formally accepted for publication in PLOS Pathogens.

Best regards,

Kasturi Haldar

Editor-in-Chief

PLOS Pathogens

orcid.org/0000-0001-5065-158X

Michael Malim

Editor-in-Chief

PLOS Pathogens

orcid.org/0000-0002-7699-2064